# Analysis of protein phosphorylation in nerve terminal reveals extensive changes in active zone proteins upon exocytosis

Mahdokht Kohansal-Nodehi[1], John JE Chua[2,3,4,5], Henning Urlaub[6,7], Reinhard Jahn[1]\*, Dominika Czernik[1]\*

[1]Department of Neurobiology, Max Planck Institute for Biophysical Chemistry, Göttingen, Germany; [2]Interactomics and Intracellular Trafficking laboratory, National University of Singapore, Singapore, Singapore; [3]Department of Physiology, National University of Singapore, Singapore, Singapore; [4]Yong Loo Lin School of Medicine, National University of Singapore, Singapore, Singapore; [5]Neurobiology/ Ageing Programme, National University of Singapore, Singapore, Singapore; [6]Bioanalytical Mass Spectrometry Group, Max Planck Institute for Biophysical Chemistry, Göttingen, Germany; [7]Bioanalytics Group, University Medical Center Göttingen, Göttingen, Germany

**Abstract** Neurotransmitter release is mediated by the fast, calcium-triggered fusion of synaptic vesicles with the presynaptic plasma membrane, followed by endocytosis and recycling of the membrane of synaptic vesicles. While many of the proteins governing these processes are known, their regulation is only beginning to be understood. Here we have applied quantitative phosphoproteomics to identify changes in phosphorylation status of presynaptic proteins in resting and stimulated nerve terminals isolated from the brains of Wistar rats. Using rigorous quantification, we identified 252 phosphosites that are either up- or downregulated upon triggering calcium-dependent exocytosis. Particularly pronounced were regulated changes of phosphosites within protein constituents of the presynaptic active zone, including bassoon, piccolo, and RIM1. Additionally, we have mapped kinases and phosphatases that are activated upon stimulation. Overall, our study provides a snapshot of phosphorylation changes associated with presynaptic activity and provides a foundation for further functional analysis of key phosphosites involved in presynaptic plasticity.

\*For correspondence: rjahn@ gwdg.de (RJ); dominika.czernik@ mpibpc.mpg.de (DC)

## Introduction

In chemical synapses, neurotransmitters are stored in synaptic vesicles and released by Ca[2+]-dependent exocytosis upon stimulation. After exocytosis, the components of the synaptic vesicle membrane are retrieved, mainly by clathrin-dependent endocytosis, and then used to regenerate synaptic vesicles for another round of exo-endocytotic recycling (**Sudhof, 2013**; **Rizzoli, 2014**).

During the past decades, the steps of the synaptic vesicle cycle have been studied in great detail. Before fusion, synaptic vesicles attach to specialized sites at the presynaptic plasma membrane (active zones), which is associated with activation (priming) of the exocytotic apparatus. Furthermore, functionally distinct pools of synaptic vesicles coexist in each nerve terminal, with only a small proportion of them participating in recycling under normal physiological conditions (**Denker et al., 2011**). Major progress has also been made in our understanding of the underlying molecular events. Synaptic vesicles contain proteins mediating neurotransmitter uptake (vesicular neurotransmitter

**eLife digest** The human nervous system contains more than a hundred billion neurons that are connected with each other via junctions called synapses. When an electrical impulse travelling along a neuron arrives at a synapse, it triggers bubble-like packages called synaptic vesicles within the neuron to merge with the neuron's surface membrane. The contents of these vesicles – chemical messengers called neurotransmitters – are then released into the synapse and carry the signal to the next neuron.

Complex molecular machines made from many different proteins control the release of neurotransmitters. Quite a few of these proteins are regulated by the addition of phosphate groups at specific sites. However, not all of the proteins involved in the release of neurotransmitters have been studied in detail and it is largely unclear how most of them are regulated.

Now, Kohansal-Nodehi et al. have used techniques involving mass spectrometry to find out which proteins have phosphate groups added or removed in neurons that are releasing neurotransmitters. The experiments used pinched-off synapses isolated from rat brains. These structures, referred to as "synaptosomes", lend themselves to this kind of study because they can be induced to continuously release neurotransmitters for several minutes. Kohansal-Nodehi et al. identified over 250 specific sites on proteins in the synaptosomes where phosphate groups are attached, including many on the key proteins known to operate in neurotransmitter release. Moreover, some proteins were modified at multiple sites, especially the proteins that form a scaffold to capture synaptic vesicles close to the membrane and prepare them for release. The data also revealed important clues about the enzymes that either attach or remove the phosphate groups.

Together, these findings provide new insights into the regulatory networks that control many proteins at the same time. The next challenge is to sort out which of these modifications change the interactions between the proteins that control neurotransmitter release, and to understand how these changes influence the trafficking of synaptic vesicles.

transporters and a vacuolar ATPase) as well as proteins involved in vesicle docking (Rab proteins), calcium sensing (synaptotagmins) and fusion (SNAREs) (*Takamori et al., 2006*). Active zones contain both scaffolding proteins such as piccolo, bassoon, liprins as well as proteins that mediate vesicle docking and priming (e.g. RIM and Munc13) (*Südhof, 2012*). Exocytotic fusion is executed by SNAREs, controlled by synaptotagmin, and regulated by additional proteins recruited from the cytoplasm such as Munc18 (*Toonen and Verhage, 2007*) and complexins (*Brose, 2008*). Clathrin-Mediated Endocytosis (CME) is mediated by a whole array of adaptor proteins, many of which containing curvature-sensing BAR and F-BAR domains (*McMahon and Boucrot, 2011*) such as dynamin that is crucial for pinching off of coated pits. Actin dynamics plays a critical role in vesicle movement and endocytosis (*Cingolani and Goda, 2008*), and additional proteins are involved in other functions such as vesicle clustering synapsins (*Cesca et al., 2010*).

Much less is known about how the synaptic vesicle cycle is regulated. Chemical synapses display profound plasticity and adaptation, and they are known to rapidly respond to a multitude of signals by changing their release properties. A large body of evidence shows that many presynaptic proteins are reversibly phosphorylated, suggesting that protein phosphorylation plays a key role in output regulation. In fact, synapsin1 was the first presynaptic protein to be discovered because it undergoes rapid phosphorylation upon depolarization of the presynaptic nerve terminal (*Ueda et al., 1981*; *Dolphin and Greengard, 1981*). Additionally, the key protein in membrane fission, dynamin, is rapidly dephosphorylated by a calcium-dependent phosphatase (calcineurin) (*Liu et al., 1994*). Dephosphorylation alters the affinity of dynamin for some of its binding partners involved in endocytosis (*Robinson et al., 1994*; *Anggono et al., 2006*; *Clayton and Cousin, 2009*). In addition to dynamin, several other proteins involved in clathrin-mediated endocytosis, termed 'dephosphins' are dephosphorylated upon depolarization, in calcium-dependent manner, including the scaffold proteins amphiphysin1 and amphiphysin2, the PtdIns(4,5)P phosphatase synaptojanin, the adapter proteins epsin, eps15, and AP180 (*Cousin and Robinson, 2001*).

Proteins mediating exocytosis also appear to be regulated by phosphorylation-dephosphorylation reactions. For example, phosphorylation of the SNARE SNAP-25 at Ser187 accelerates vesicle recruitment (*Nagy et al., 2002*), regulates formation of SNARE complex (*Gao et al., 2016*) and consequently promotes exocytosis in PC12 cells, and it is also induced by stress in affected brain regions (*Yamamori et al., 2014*). Additionally, it was reported that phosphorylation of SNAP-25 at Thr138 interferes with assembly of SNARE complex and exocytosis (*Gao et al., 2016*). Moreover, phosphorylation of syntaxin1 at Ser14 and Ser188 regulates its interaction with Munc18-1 (*Tian et al., 2003*; *Rickman and Duncan, 2010*). Dephosphorylation of N-ethylmaleimide-sensitive factor (NSF) at Tyr 83 and Thr645 was reported to promote secretory vesicle fusion (*Huynh et al., 2004*), probably by enhancing disassembly of SNARE complexes (*Belluzzi et al., 2016*).

Certain presynaptic proteins have multiple phosphorylation sites that can simultaneously undergo opposing changes in their phosphorylation states. Best characterized are the synapsins that are phosphorylated at multiple sites by several different kinases. For example, synapsin1 is phosphorylated at Ser9 by $Ca^{2+}$/calmodulin-dependent protein kinase I (CaMKI). It is also phosphorylated by mitogen-activated protein kinase (MAPK) and CdK5 at both N and C termini (Ser62, Ser67, Ser549 and Ser551) and by CaMKII at its C terminus (Ser566 and 603) (*Cesca et al., 2010*). Moreover, the tyrosine-kinase Src phosphorylates synapsin1 at Tyr301 (*Onofri et al., 2007*). Intriguingly, the functional consequences of synapsin phosphorylation/dephosphorylation at various phosphosites are different. While phosphorylation of Ser9, Ser566 and Ser603 upon stimulation decreases actin binding and increases exocytosis of synaptic vesicles, phosphorylation of Tyr301 has the opposite effect (*Cesca et al., 2010*). Moreover, phosphorylation at Ser62, Ser67, Ser549 and Ser551 is downregulated upon stimulation of neurotransmitter release and decreases the binding of synapsin to actin filaments, leading to the suggestion that phosphorylation by Cdk5 at Ser549 and 551 defines the ratio between resting and recycling synaptic vesicle pools (*Verstegen et al., 2014*).

In the past, synaptic phosphoproteins were mostly identified by *in vitro* assays using cell/tissue extracts or purified proteins. Based on this classical approach, we know that for example syntaxin1A, synaptobrevin (VAMP) and SNAP25 are phosphorylated by calcium/calmodulin dependent protein kinase type II (CaMKII); synaptobrevin and SNAP25 by protein kinase C (PKC), and synaptotagmin1 and syntaxin1A by casein kinase II (CK2) (*Bennett et al., 1993*; *Nielander et al., 1995*; *Hirling and Scheller, 1996*; *Shimazaki et al., 1996*).

While these classical approaches typically focus on individual proteins, modern screening methods have considerably enlarged the coverage of phosphoproteins. The introduction of chip arrays and synthetic peptides as substrates for kinases has allowed the mapping of consensus motifs for individual kinases (*Ptacek et al., 2005*), which were later used to develop algorithms for predicting phosphosites *in silico* (*Linding et al., 2008*). Commercially available kinase libraries combined with *in vitro* synthesis of peptides permit extended screening of predicted phosphorylation sites although it needs to be borne in mind that this method is prone to both false negatives and false positives. Most importantly, advanced mass spectrometry (MS) methodologies have completely changed the field as they permit the identification of phosphorylation sites in a robust, global and quantitative manner (*Huttlin et al., 2010*; *de Graaf et al., 2014*; *Sharma et al., 2014*; *Giansanti et al., 2015*). Over the last few years, progress in MS-based techniques for phosphopeptide identification and quantification has allowed global and in-depth analysis of thousands of phosphosites in a single analytical run (*Lemeer and Heck, 2009*; *Kelstrup et al., 2014*; *Ludwig et al., 2015*). These novel techniques have triggered a rapid increase in studies dealing with global phosphoproteome maps of tissues and organs, resulting, for instance, in an atlas of mouse phosphorylations (*Huttlin et al., 2010*) and quantitative maps of phosphorylations in rat organs (*Lundby et al., 2012*).

Several studies have dealt with global phosphoproteomics in the brain (*Ballif et al., 2004*; *2008*; *Trinidad et al., 2008*; *Tweedie-Cullen et al., 2009*; *Palmisano et al., 2012*; *Goswami et al., 2012*; *Corradini et al., 2014*; *Tagawa et al., 2015*) or in isolated nerve terminals (synaptosomes) (*Munton et al., 2007*; *Collins et al., 2008*; *Filiou et al., 2010*; *Trinidad et al., 2012*). Nevertheless, we still do not have a systematic overview of phosphorylation changes in the presynaptic terminal that are associated with the triggering of neurotransmitter release.

To close this gap, we have investigated changes in the presynaptic phosphoproteome during stimulation of exocytosis by taking advantage of synaptosomes that constitute pinched off and resealed nerve terminals enriched from brain homogenate. Synaptosomes retain the ability to synthesize ATP, maintain the resting potential and respond to depolarization with $Ca^{2+}$-influx and $Ca^{2+}$-

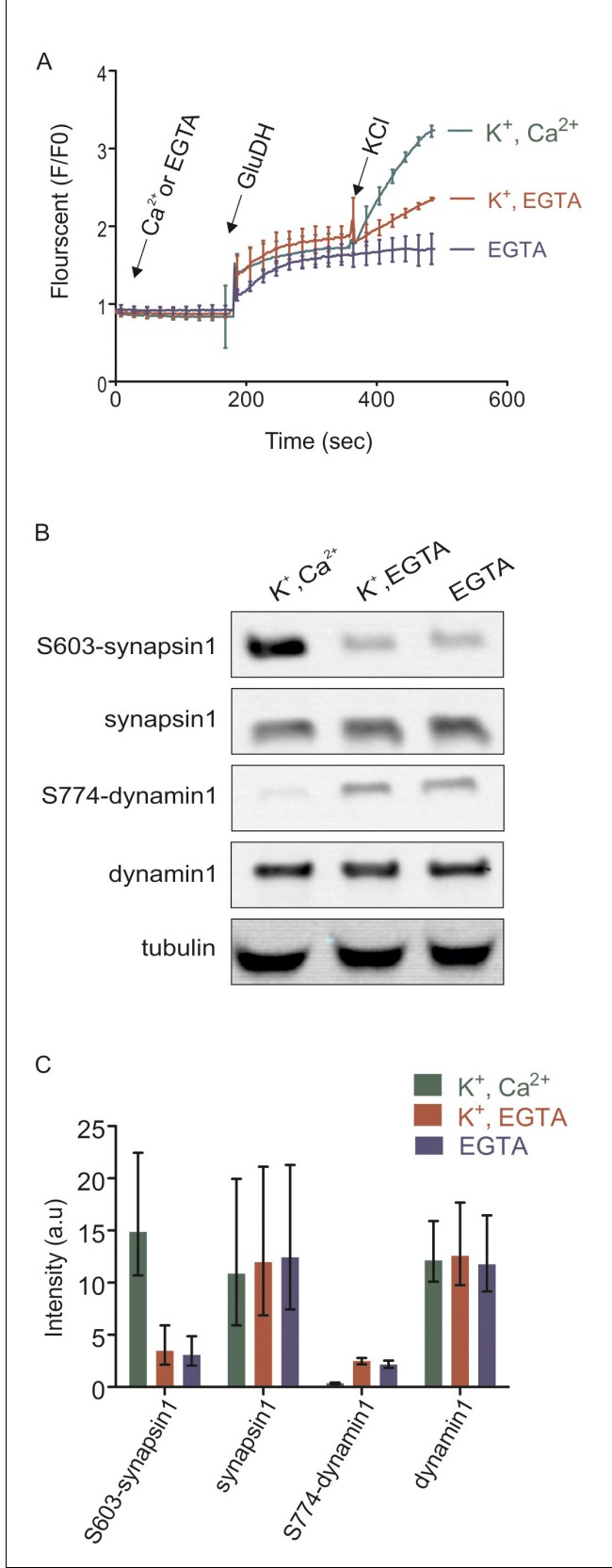

**Figure 1.** Isolated synaptosomes are responsive to stimulations. (**A**) Glutamate release by synaptosomes, monitored by enzymatic conversion of glutamate by glutamate dehydrogenase (GluDH) (see Materials and

*Figure 1 continued on next page*

*Figure 1 continued*

methods). The functionality of isolated synaptosomes was also assessed by immunoblot detection of phosphorylated Ser603 of synapsin1 and Ser774 of dynamin1 (**B** and **C**). (**A**) Synaptosomes were preincubated in buffer containing either $Ca^{2+}$ (final concentration 1.3 mM) or EGTA (final concentration 0.5 mM), followed by sequential addition of GluDH (200 U) and KCl ($K^+$) for depolarization (final concentration 50 mM) or not stimulated (no $K^+$ was added). Data represented as mean of three replicates, with the bars indicating the range of values. (**B** and **C**) Immunoblot analyses of phosphorylation changes in synapsin1 (Ser603) and dynamin1 (Ser774). Synaptosomes monitored and treated 2 min after addition of KCl as described in (**A**) were analyzed by immunoblotting using phosphospecific antibodies. (**B**) Representative immunoblots. To ensure equal loading, all samples were also blotted using antibodies insensitive to phosphorylation change, and tubulin as a loading control. (**C**) Quantification of the blot signals obtained from three independent experiments shown as the mean of the three replicates, with the error bars indicating the range of values.

dependent exocytosis of synaptic vesicles (*Nicholls, 2003*). Synaptosomes respond not only to external signals but also to activators and inhibitors of various steps of the synaptic vesicle cycle such as clostridial neurotoxins (*Blasi et al., 1994*), black widow spider venom (*Nicholls et al., 1982*), or dynamin inhibitors such as dynasore (*Daniel et al., 2012*). Indeed, synaptosomes were used to monitor stimulation-dependent changes in the phosphorylation of individual proteins such as synapsin and dynamin. Here we have used synaptosomes as a tool for the study of the presynaptic phosphoproteome. We observed many hitherto unknown changes in the phosphorylation pattern of, in particular, active zone proteins that exhibit concurrent changes in phosphorylation and dephosphorylation at multiple locations/positions. Our results show that protein phosphorylation plays an even more important role than previously anticipated and appears to be critically involved in regulating active zone function.

## Results

### Phosphoproteomics of isolated nerve terminals under different stimulation conditions

For the analysis of the presynaptic phosphoproteome, we prepared isolated nerve terminals (synaptosomes) from brain cortices of 5–6 weeks old rats using standard subcellular fractionation procedures (see Materials and methods). To examine whether the synaptosomes are responsive to stimulation, we performed two independent quality checks. First, we measured glutamate release upon depolarization by increasing the external potassium concentration. While this treatment clamp-depolarizes the membrane, calcium-overload is prevented by inactivation of the calcium channels (*Simms and Zamponi, 2014*).

As expected, addition of KCl led to a fast increase in glutamate release that persisted for several minutes after depolarization and that was reduced in the absence of $Ca^{2+}$ (*Figure 1A*). It has been shown that the difference between glutamate release in the presence and absence of calcium represents exocytotic release whereas the increase upon depolarization in the absence of $Ca^{2+}$ is due to non-exocytotic release caused by the reversal of glutamate transporters in the plasma membrane that depend on the membrane potential (*Nicholls and Sirha, 1986*). Second, we monitored the phosphorylation status of two well-characterized phosphosites in the nerve terminal, synapsin1 (Ser603) and dynamin1 (Ser774), using phosphospecific antibodies. These two sites are known to be regulated antagonistically by $Ca^{2+}$-influx, with synapsin being phosphorylated and dynamin being dephosphorylated (*Jovanovic et al., 2001*; *Graham et al., 2007*). As shown in *Figure 1B and C*, both sites changed their phosphorylation status as expected, confirming that the synaptosomes are functional. Both tests for glutamate release and alteration of marker phosphorylation sites were carried out for all preparations before mass spectrometry (LC-MS/MS) analysis.

For phosphoproteomic analysis, we compared three different incubation conditions of synaptosomes: (i) depolarization with ($K^+$, $Ca^{2+}$) or (ii) without calcium ($K^+$, EGTA), and (iii) control (no depolarization, EGTA). Stimulation of all samples were stopped after 2 min by the addition of ice-cold lysis buffer and processed as described in Materials and methods. While rapid and reversible phosphorylation changes may not be detectable anymore at this time point, both exo- and endocytosis

are at an activated steady-state level. Equal amounts of protein were digested by trypsin. For quantification, we used stable isotope dimethyl labeling (*Boersema et al., 2009*). To elucidate the changes in phosphorylation status of phosphosites, we compared the phosphoproteome of depolarized synaptosomes in the presence of $Ca^{2+}$ against depolarized synaptosomes in the absence of $Ca^{2+}$ and synaptosomes that were not depolarized. In each set of comparisons, one of the incubation conditions was labeled as heavy and the other one as light. The differentially labeled peptides from two conditions were then mixed, and fractionated by strong cation exchange (SCX) chromatography. This step was introduced since it reduced sample complexity and resulted in higher recovery rates of phosphopeptides (data not shown). In the next step, each fraction obtained from the SCX-column was separately enriched for phosphopeptides, and analyzed by MS (see Materials and methods for details and *Figure 2—figure supplement 1* for an overview of the workflow). *Figure 2—figure supplement 5* shows exemplary MS spectra obtained for phosphopeptides of synapsin (Ser603) and dynamin (S774) after mixing heavy and light labeled samples, and the corresponding MS/MS spectra used for sequence determination.

In parallel, we determined the entire proteome of synaptosomes (three biological replicates) using standard procedures (see Materials and methods for details). We identified 4961 proteins (for list of identified proteins see Supplementary file 1, available at Dryad [*Kohansal-Nodehi et al., 2016*]) in the proteome and 1257 phosphoproteins in the phosphoproteome. We compared the biological functions enriched in our proteome and phosphoproteome datasets using Ingenuity Pathway Analysis (IPA, http://www.ingenuity.com). The analysis revealed that most of the functions related to neurotransmission and synaptic vesicle recycling are enriched in both proteomes, with a higher enrichment being observed in the phosphoproteome compared to the proteome (*Figure 2—figure supplement 2*). Note that due to redundancies and multiple overlaps in the functional categories, this type of analysis constitutes only a rough estimate. However, it documents that even though synaptosomes are contaminated with many other organelles and post synaptic density proteins, the phosphorylation events occur predominantly, and perhaps exclusively, inside the presynaptic nerve terminal under our experimental conditions.

To quantify changes of phosphorylation sites under different incubation conditions, we performed pairwise comparisons of the phosphoproteome in the respective synaptosome samples. All data are based on three biological replicates. Phosphosites that were quantified in at least two out of three replicates were filtered and subjected to downstream data analysis to determine whether they are differentially (up or down) regulated (see Materials and methods). Due to the rigorous selection criteria, the coverage of phosphosites that were reliably quantified in the depolarization with $Ca^{2+}$ versus depolarization without $Ca^{2+}$ comparison and depolarization with $Ca^{2+}$ versus no depolarization comparison corresponded, respectively, to 67.9% and 62.5% of the total phosphosites that were initially identified. The complete list of quantified phosphosites in all three comparisons is reported in Supplementary file 2, available at Dryad (*Kohansal-Nodehi et al., 2016*). To assess the reproducibility of the biological replicates, the Pearson correlation coefficients between dimethyl ratios and intensities of phosphosites in three biological replicates were determined (*Figure 2—figure supplement 3*). High correlation between the replicates in both comparison were confirmed, with the coefficients greater than 0.72 for intensities and greater than 0.65 for dimethyl ratios (*Figure 2—figure supplement 3A–D*).

The overall results of the comparative quantitative analysis are shown in *Figure 2*. Several important conclusions can be drawn from this analysis. First, only a small fraction (12–15% ) of all quantified phosphosites show significant changes upon depolarization in the presence of $Ca^{2+}$ (*Figure 2A–C*). Both phosphorylation and dephosphorylation events are detectable, with phosphorylation events being slightly more prominent (*Figure 2C*). The overlap among all quantified phosphosites is very high between the two comparisons (*Figure 2D*). There is also substantial overlap in regulated phosphosites between the comparisons (37% in upregulated and 35% in downregulated phosphosites, *Figure 2E–G*) with identical tendency of regulation and similar fold changes (*Figure 2H*). Taken together, these data show that only a limited set of proteins undergo rapid phosphorylation changes that are specifically triggered by $Ca^{2+}$, most probably involving $Ca^{2+}$-dependent kinases and phosphatases (see below). As expected, under stimulation conditions, synapsin1 (Ser603) and dynamin1 (Ser774) (*Figure 1A*) were identified as up- and downregulated phosphosites, respectively (*Figure 2A and B*). This result agrees with our western blot analyses using phospho-specific antibodies, confirming the high quality of the datasets.

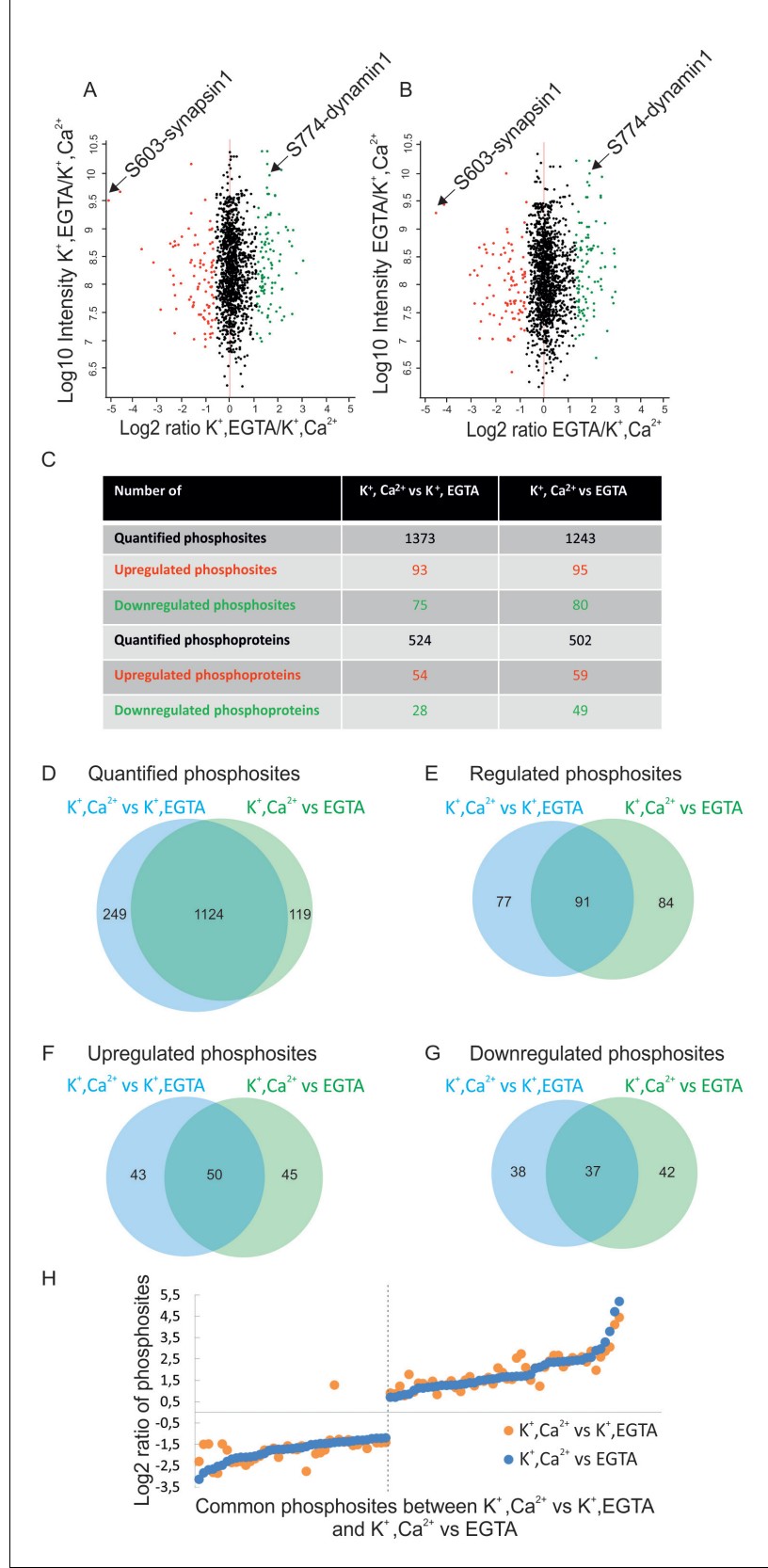

**Figure 2.** Overview of the quantified phosphosites identified in resting and stimulated synaptosomes. Distribution of quantified phosphosites from pair-wise comparisons of (**A**) depolarization with Ca$^{2+}$ versus depolarization

*Figure 2 continued on next page*

*Figure 2 continued*

without $Ca^{2+}$ ($K^+$, $Ca^{2+}$ versus $K^+$, EGTA) (**B**) depolarization with $Ca^{2+}$ versus no depolarization ($K^+$, $Ca^{2+}$ versus EGTA). Quantifications were done by dimethyl labeling and significant outliers were determined using the Perseus module based on two-tailed 'Significant A' test (p-value ≤0.05) (see Materials and methods). Red dots represent upregulated phosphosites, green dots represent downregulated sites when comparing depolarization in the presence to absence of $Ca^{2+}$. Positions represented by black dots did not change (**C**) Number of quantified, up- or down-regulated phosphosites and phosphoproteins in the two comparisons. (**D–G**) The proportional Venn diagrams show the overlap of quantified phosphosites (**D**), regulated phosphosites (**E**), upregulated phosphosites (**F**) and downregulated phosphosites (**G**) between the two conditions indicated above. (**H**) Plot comparing the extent of stimulation-dependent phosphorylation changes with respect to non-depolarized and depolarized control samples. All regulated phosphosites shared between $K^+$, $Ca^{2+}$ versus $K^+$, EGTA and $K^+$, $Ca^{2+}$ versus EGTA comparisons are depicted.

The following figure supplements are available for figure 2:

**Figure supplement 1.** Workflow for large-scale quantitative phosphoproteomics of synaptosomes.

**Figure supplement 2.** Comparison of functional categories for proteins identified in the phosphoproteome versus the proteome of synaptosomes.

**Figure supplement 3.** Correlation of phosphoproteome between biological replicates.

**Figure supplement 4.** Phosphorylation status and its change upon membrane depolarization.

**Figure supplement 5.** MS1 and MS/MS spectrum of synapsin Ser603 and dynamin Ser774.

To check whether depolarization alone leads to changes in the phosphorylation state of proteins, we compared the phosphoproteome of depolarized with non-depolarized synaptosomes in the presence of the calcium chelator EGTA. Although some statistically significant changes were observed, they were less pronounced and clustered close to the threshold (compare *Figure 2A and B* with *Figure 2—figure supplement 4A*). Moreover, the correlation of the ratios was low (*Figure 2—figure supplement 4C*) although the Pearson correlation of the intensities was high (*Figure 2—figure supplement 4B*). For these reasons, this comparison was not considered in our further analysis.

## Activity-dependent phosphorylation changes in presynaptic proteins

Given that the large number of phosphosites detected were unchanged in the relatively large number of quantified phosphorylation sites, the corresponding small fraction of sites that underwent stimulation dependent changes is highly significant and potentially of importance in regulating neurotransmitter release. To gain an overview about the functional properties of the proteins exhibiting phosphosite changes in the two comparisons, we annotated them manually and classified into 15 groups based on their function and localization (*Figure 3*). With only a few exceptions (e.g. translation/transcription or neurotransmitter metabolism related proteins), proteins undergoing phosphorylation changes during $Ca^{2+}$-dependent stimulation are all involved in synaptic vesicle trafficking or in presynaptic signaling (*Figure 3A and B*), again documenting the specificity of the experimental approach. Note that postsynaptic proteins, particularly proteins associated with the post synaptic density, were largely missing (*Figure 3A and B*).

A more detailed representation of the quantified phosphosites is shown in *Figure 4*. Here, the size of the circles is proportional to the total number of quantified phosphosites in the two comparisons. Red, green and white areas in the circles are proportional to the number of up, down and non-regulated phosphosites, respectively. The lines connecting the proteins show the interaction of the proteins based on STRING database and literature mining (Active Zone (AZ) proteins (*Ohtsuka et al., 2002*; *Schoch et al., 2002*; *Wang et al., 2002*; *Takao-Rikitsu et al., 2004*; *Lu et al., 2005*; *Schoch and Gundelfinger, 2006*; *Wang et al., 2009*), Clathrin Mediated Endocytosis (CME) proteins (*Ringstad et al., 1997*; *Anggono et al., 2006*; *Anggono and Robinson, 2007*)). The figure shows that protein groups involved in synaptic vesicle recycling that exhibit stimulation-dependent changes in phosphorylation include active zone proteins, synaptic vesicle proteins,

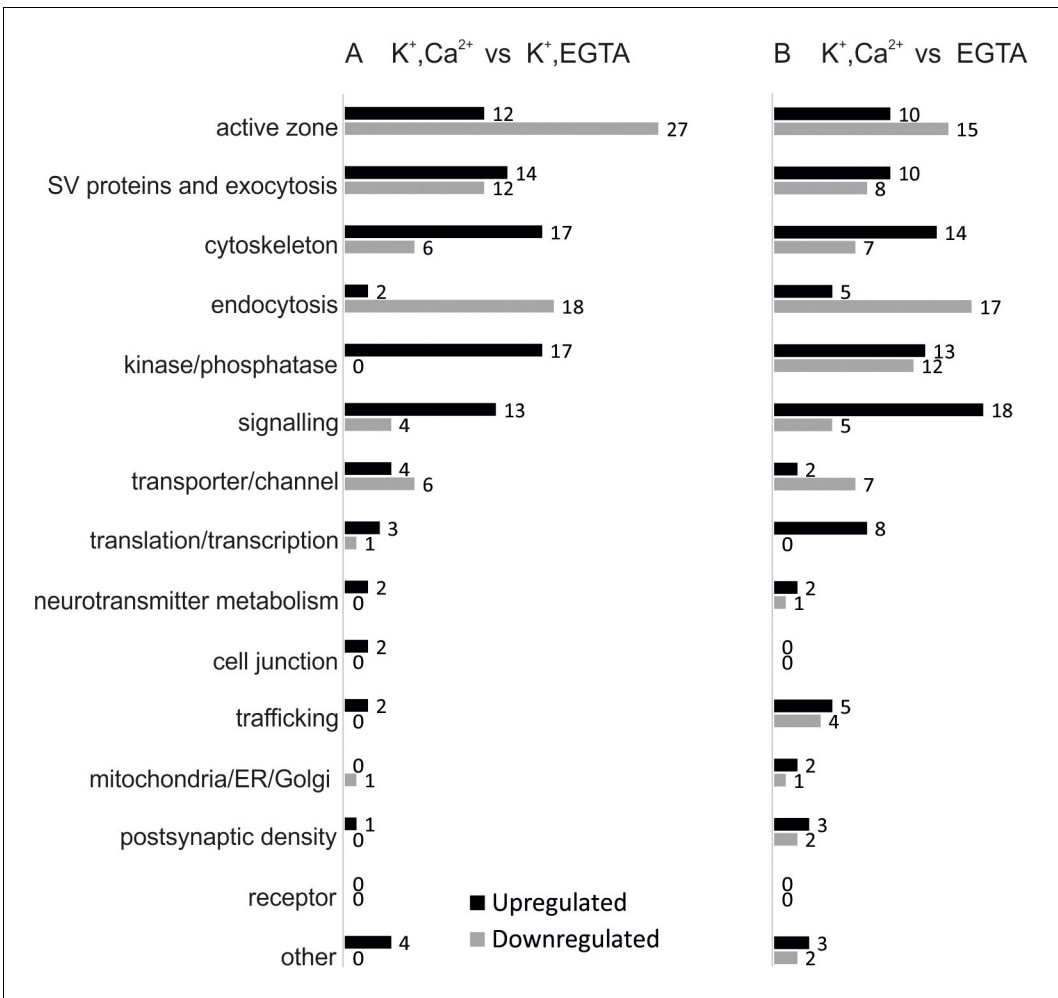

**Figure 3.** Overview of the regulated phosphosites, based on corresponding grouping of protein functions. Proteins containing regulated phosphosites were individually divided into 15 groups as described previously (**Boyken et al., 2013**). Proteins with function/localization other than listed or uncharacterized proteins were grouped into the 'other' category.

endocytotic proteins, and few proteins in the presynaptic plasma membrane (mainly ion channels). In addition, it is evident that some of the proteins (e.g. bassoon, RIM1, synapsin1, dynamin1) contain multiple phosphorylation sites that can undergo parallel phosphorylation and dephosphorylations. Surprisingly, proteins of the active zone were most heavily phosphorylated, which is discussed further below.

With few exceptions, most of the phosphosites in the Clathrin-Mediated Endocytosis (CME) functional group were dephosphorylated upon stimulation in the presence of $Ca^{2+}$ (13 out of 15 regulated phosphosites in the depolarization with $Ca^{2+}$ versus depolarization without $Ca^{2+}$ comparison and 12 out 17 in the depolarization with $Ca^{2+}$ versus no depolarization comparison) (**Figure 4A and B**). This is in agreement with the previous report (**Cousin and Robinson, 2001**). Phosphorylation/ dephosphorylation of CME proteins was proposed to serve two related functions. First, phosphorylation regulates the interactions between different CME proteins. For example, phosphorylation of the adaptor protein AP180 weakens its interaction with the AP-2 adaptor complex, which is required for synaptic vesicle endocytosis (**Hao et al., 1999**). In our dataset, three AP180 phosphosites (Ser600, Ser621 and Ser627) are dephosphorylated, one of which (Ser627) is located in the reported binding site. Second, phosphorylation generally weakens the affinity towards membranes containing acidic lipids and thus promotes dissociation of coat proteins. For instance, phosphorylation of dynamin1 by PKC at Ser795 prevents its association with phospholipids (**Powell et al., 2000**).

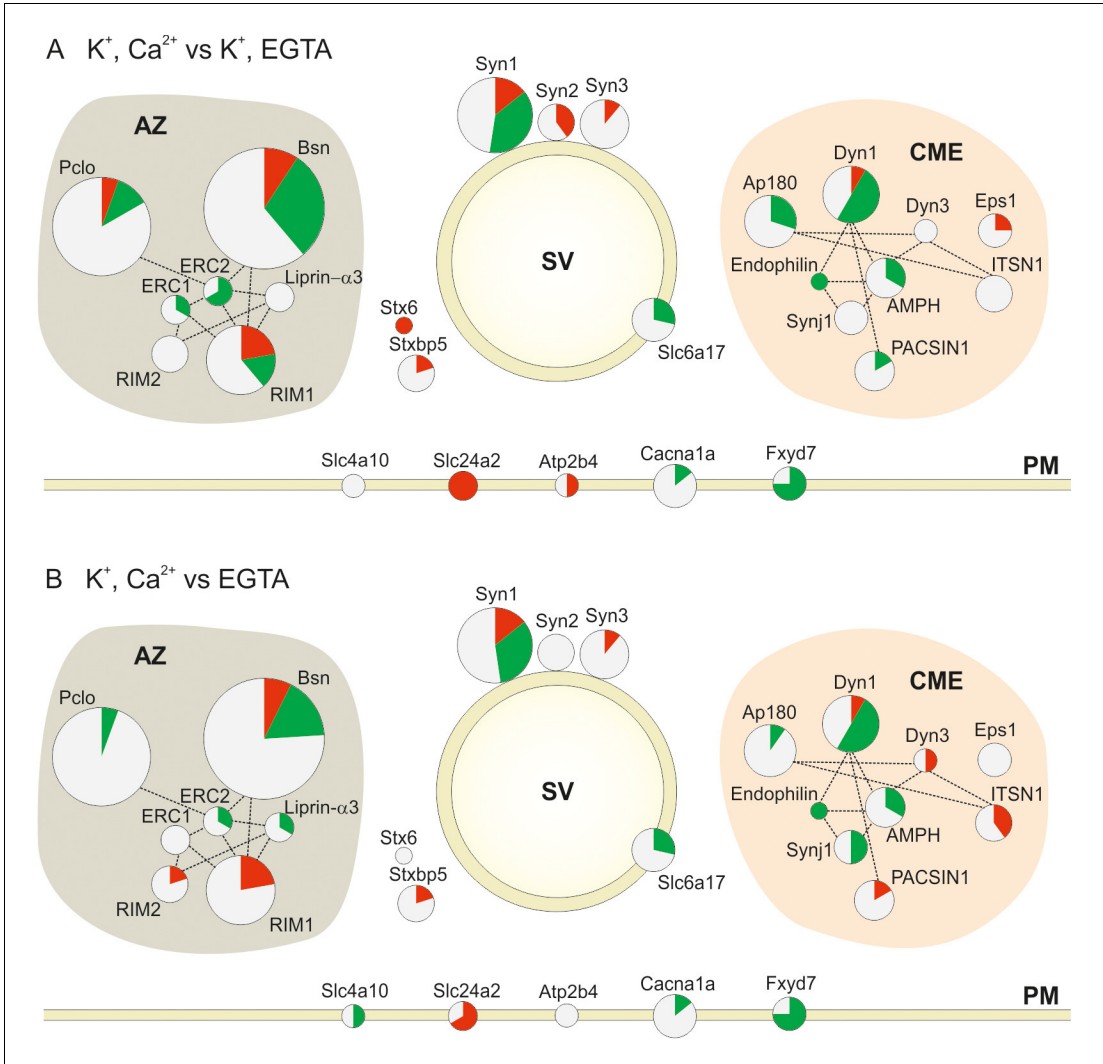

**Figure 4.** Changes in the phosphorylation status of proteins involved in synaptic vesicle trafficking following stimulation. Overview of changes in the phosphorylation status of Active Zone proteins (AZ), synaptic vesicle proteins (SV), proteins involved in Clathrin Mediated Endocytosis (CME) and Plasma Membrane proteins (PM) in the two pair-wise comparisons: (**A**) depolarization with $Ca^{2+}$ versus depolarization without $Ca^{2+}$ ($K^+$, $Ca^{2+}$ versus $K^+$, EGTA) (**B**) depolarization with $Ca^{2+}$ versus no depolarization ($K^+$, $Ca^{2+}$ versus EGTA). The circles are proportional to the number of quantified phosphosites in the two conditions. The red, green and white area in each circle is proportional to the number of up, down and nonregulated phosphosites, respectively. The dotted lines connecting the circles show the interactions of the proteins based on STRING database and literature mining (references included in the main text; for a glossary of the abbreviations see supplemental *Figure 4—source data 1*.

The following source data is available for figure 4:

**Source data 1.** Protein name, Gene name, Uniprot ID and abbreviation of proteins shown in *Figure 4*.

*Figure 4* also confirms and extends previous reports on the regulation of multiple phosphosites in synapsin1 and dynamin1. For synapsin1, regulation of some of the sites has been previously characterized extensively (*Jovanovic et al., 2001*). Such as downregulation of Ser62 and upregulation of Ser556 and Ser603. However, no stimulus-dependent changes were observable at the phosphorylation sites of CdK5 (Ser549 and 551) that were suggested to regulate the size of the synaptic vesicle pools (*Verstegen et al., 2014*).

Components of the presynaptic active zone make up the most conspicuously phosphorylated proteins (e.g. bassoon, piccolo, RIM1, liprin-α3 and ERC1-2). The amino acid positions of phosphosites identified on these proteins are shown in *Figure 5*, with the well characterized phosphoproteins synapsin1 and dynamin1 included for comparison. Similarly to synapsin and dynamin, we observed

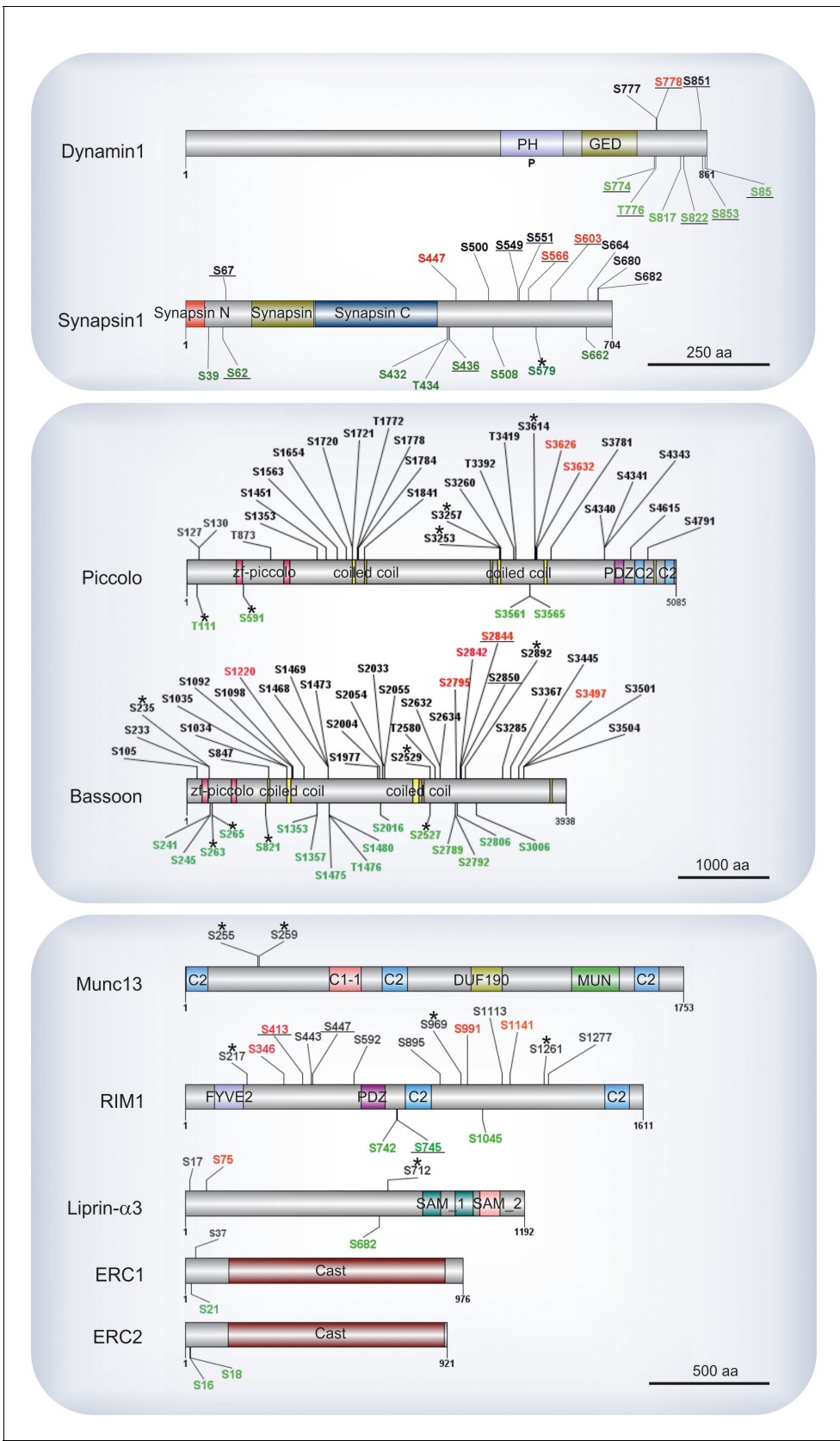

**Figure 5.** Phosphosites identified in proteins of the active zone. Colors indicate phosphosites that are upregulated (red), downregulated (green), or remain unchanged (black) in depolarization with $Ca^{2+}$ versus depolarization without $Ca^{2+}$ comparison ($K^+$, $Ca^{2+}$ versus $K^+$, EGTA). The underlined

*Figure 5 continued*

phosphosites represent sites previously reported to be physiologically relevant (for references see Supplementary file 3, available at Dryad [**Kohansal-Nodehi et al., 2016**]). Unmarked phosphosites represent sites previously reported only by proteomic discovery-mode mass spectrometry, defined as HTP in the publicly available PhosphoSitePlus database. Phosphosites marked with a star represent phosphosites reported here for the first time. For comparison, phosphosites found in the well-studied proteins dynamin1 and synapsin1 are shown.

many sites of active zone proteins undergoing parallel changes in phosphorylation and dephosphorylation upon $Ca^{2+}$-dependent stimulation. Active zone proteins regulate synaptic transmission speed, precision, and plasticity (*Kaeser, 2011*; *Sudhof, 2012*). Our results show that, the proteins bassoon and piccolo are massively phosphorylated, with 48 and 31 sites being quantified in our experiments, respectively. Intriguingly, with the exception of 6 phosphosites, most of the piccolo phosphosites remained unchanged following stimulation. In comparison, bassoon exhibited a dramatic decrease in phosphorylation at multiple sites in addition to increased phosphorylation at 5 other positions. While the role of these novel sites is still unclear, it is conceivable that downregulation of the bassoon phosphosites may serve to regulate the global properties of the protein (e.g. surface charge density) rather than specific protein-protein interactions. In previous studies it has been reported that in spite of having similar structural motifs, bassoon and piccolo have different protein interaction partners (*Takao-Rikitsu et al., 2004*; *Schoch and Gundelfinger, 2006*; *Wang et al., 2009*). The more pronounced observed dephosphorylation of phosphosites in bassoon compared to piccolo can be an explanation for their different protein interactions.

RIM, considered to be central organizer of active zone, is known to have several interactions with other active zone proteins as well as with calcium channels and SV proteins such as Rab3a (*Südhof, 2012*). Here, we identified 7 regulated phosphosites that are distributed along the protein (*Figure 5*) including Ser413 that was previously reported to be phosphorylated by PKA, and its role in presynaptic long-term plasticity has been discussed extensively (*Lonart et al., 2003*; *Kaeser et al., 2008*). A summary of all previously characterized phosphosites that were assigned as significantly regulated phosphosites in our datasets is given in Supplementary file 3, available at Dryad (*Kohansal-Nodehi et al., 2016*).

## Calcium-dependent kinases and phosphatases are mainly responsible for phosphorylation changes after stimulation.

Next, we tried to find out which kinases and phosphatases are responsible for the phosphorylation changes upon stimulation. First, we analyzed the respective sites for sequence motifs known to be targeted by specific kinases. It is well established that the substrate specificity of Ser/Thr kinases is strongly influenced by the amino acids surrounding the phosphorylated Ser/Thr residues, with specific consensus motifs being described for an array of protein kinases (*Ubersax and Ferrell, 2007*). To establish whether there are over-represented consensus sites, the phosphorylation sites were analyzed by the Motif-X tool (*Schwartz and Gygi, 2005*). Phosphosites upregulated during $Ca^{2+}$-dependent stimulation were highly enriched for the RxxS sequence motif (7.36 fold, p=1.0 × 10$^{-16}$ compared to control and 5.98 fold, p=4.8 × 10$^{-13}$, when compared to samples depolarized in the absence of $Ca^{2+}$)(*Figure 6A and B*). RxxS is the consensus motif of calmodulin-dependent kinase II (CaMKII) but also targeted by protein kinase A (PKA) and conventional isoforms of protein kinase C (PKC). This is in agreement with previous studies showing that these kinases are activated upon depolarization of nerve terminals in the presence of $Ca^{2+}$ ions (*Millán et al., 2003*; *Yamauchi, 2005*).

A similar analysis using the Motif-X tool was also carried out for downregulated phosphosites. Here, SP sequence motif was enriched significantly (8.21 fold, p=1.0 × 10$^{-16}$ when compared to control and 8.03 fold, p=1.0 x 10$^{-16}$ when compared to samples depolarized in the absence of $Ca^{2+}$-*Figure 6C and D*, respectively). Phosphatases are not known to have preferred substrate consensus motifs (*Kennelly and Krebs, 1991*). However, the SP motif is preferred by proline-directed kinases such as ERK1/2, GSK-3 and Cdk5 (*Guan et al., 1991*; *Lee et al., 1997*; *Veeranna et al., 1998*). Between the mentioned kinases, ERK1/2 is less likely to be inactivated upon stimulation, since we have quantified the upregulation of two activating phosphosites in the active loop of ERK1/2 (T183, Y185, Supplementary file 2, K$^+$, EGTA vs K$^+$, $Ca^{2+}$).

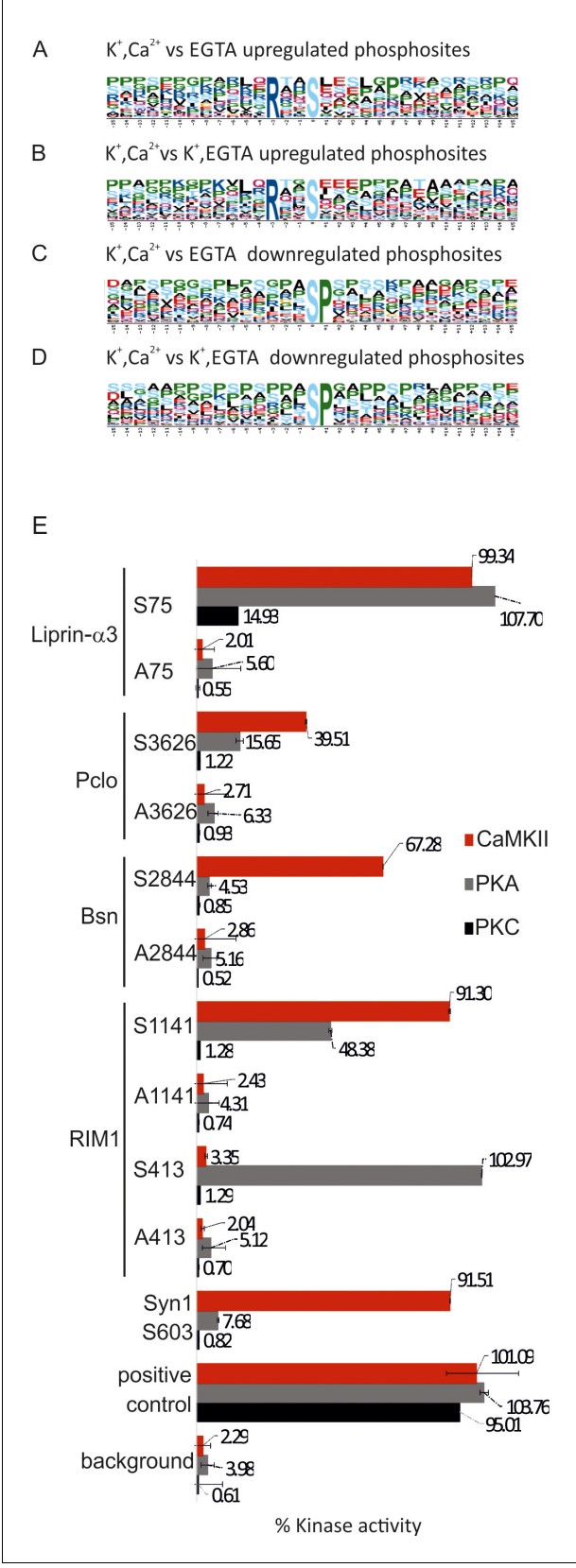

**Figure 6.** Analysis of sequence motifs and experimental verification of specific kinase phosphorylation on selected phosphosites. (**A–D**) Sequence analysis of regulated phosphosites using the Motif-X tool (*Schwartz and Gygi,*

*Figure 6 continued*

*2005*). The frequency of residues surrounding the phosphorylated serine is indicated by the size of the letters. Among the upregulated sites, the motif RXXS is overrepresented, whereas the SP motif is conspicuously present among downregulated motifs. (**E**) *In vitro* kinase assays evaluating the ability of selected kinases to phosphorylated phosphosites identified on active zone proteins. Five phosphosites were selected from four active zone proteins for testing against CaMKII, PKA, and PKC. For each site, two 13-mer peptides were synthesized as substrates (for sequences of the peptide see *Figure 6—source data 1*) that correspond to the sequence surrounding the phosphorylated serine, with one of them containing an alanine instead of the serine as control. The following peptides were used as positive controls: (i) synapsin S603 that is known to be phosphorylated by CaMKII (*Jovanovic et al., 2001*) and (ii) autocamide-2, kemptide, and neurogranin were used as known substrates of CaMKII, PKA, and PKC, respectively. Background was determined by omitting peptide from the assay. The data show means of three replicates, with the bars indicating the range of values.

The following source data is available for figure 6:

**Source data 1.** Sequence of the peptides used in *in vitro* kinase assay.

---

Overall, it is conceivable that phosphosites with SP motifs turnover more rapidly, with the decrease reflecting either reduced kinase activity or increased processing by the phosphatases. Net dephosphorylation of SP-sites may have functional consequences for presynaptic function. For instance, CdK5 is known to be major control point of neurotransmitter release (*Kim and Ryan, 2010*). The balance between the activation of calcineurin (PP2B) and Cdk5 is reported to control the dynamics of resting and recycling pools as well as activity of voltage gated calcium channels (*Kim and Ryan, 2013*). Also the coordination of CdK5 activity and other proline directed kinases such GSK-3 and ERK1/2 can result in regulation of neurotransmitter release. For example, Cdk5 and GSK-3 phosphorylate P/Q- and N-type voltage-dependent calcium channels (*Tomizawa et al., 2002*; *Zhu et al., 2010*) whereas ERK1/2 phosphorylate L-type calcium channels (*Subramanian and Morozov, 2011*) and phosphorylation decreases the activity of both types of calcium channels. Thus, it is conceivable that $Ca^{2+}$ influx increases the activity of calcium channels via dephosphorylation. However regulation of calcium channels via $Ca^{2+}$ influx seems to be more complex since they are also inactivated by calcium dependent mechanisms (*Simms and Zamponi, 2014*).

Due to the lack of consensus motifs it is not possible to identify the phosphatases that are activated upon stimulation using sequence analysis. However, the increase in the $Ca^{2+}$ concentration activates phosphatase PP2B (calcineurin) (*Sun et al., 2010*). Also, in our dataset we observe that the regulatory subunit delta (LOC100909464) of protein phosphatase 2A shows increased phosphorylation at Ser567 (Supplementary file 2, $K^+$, EGTA vs $K^+$, $Ca^{2+}$, *Kohansal-Nodehi et al., 2016*), a site known to be phosphorylated by PKA, resulting in an increase of its enzymatic activity (*Ahn et al., 2007*). This suggests the activation of PP2A upon stimulation. We also observe phosphorylation changes in Neurabin-2 (S100), regulatory protein of protein phosphatase 1 (PP1), (Supplementary file 2, $K^+$, EGTA vs $K^+$, $Ca^{2+}$). It has been shown previously that phosphorylation of S100 alters PP1 localization by the decrease of binding affinity to actin and the diffusion from the nerve terminal (*Colbran, 2004*). In summary, our data suggest that upon $Ca^{2+}$ influx subset of kinases and phosphatases such as PKA, PKC, CaMKII, PP2A and PP2B are activated while PP1 is less active and the target sites of proline directed kinases show a decrease in phosphorylation, mediated either by decreased activity of the kinases or preferred targeting by the phosphatases.

To confirm that hitherto uncharacterized phosphosites containing the RxxS motif are indeed phosphorylated by one or more of the three kinases known to target these sites, five of these sites were selected for experimental verification: RIM1 (Ser413 and Ser1141), bassoon (Ser2844), piccolo (Ser3626) and liprin-α3 (Ser75). For each of these sites, 13-mer peptides were synthesized (containing 6 residues upstream and 6 residues downstream of the phosphoserine) and tested by a commercially available kinase assay against CaMKII, protein kinase A, and protein kinase C. As a control, corresponding peptides were synthesized, in which the phosphorylated serine was replaced by alanine. As shown in *Figure 6E*, each of the peptides was recognized by at least one of the kinases. For instance, RIM1 Ser413 was phosphorylated only by PKA; consistent with a previous report (*Lonart et al., 2003*). RIM1 Ser1141 was phosphorylated mostly by CaMKII and with lower efficiency by PKA. Bassoon Ser2844 was phosphorylated only by CaMKII. Piccolo Ser3626 was phosphorylated

by CaMKII and PKA albeit with low efficiency. Liprin-α3 Ser75 was the only phosphosite that was phosphorylated by all three kinases although PKC was less efficient. Together, these findings agree with the notion that activation of these three kinases during depolarization in the presence of $Ca^{2+}$ leads to the phosphorylation of the active zone proteins at their corresponding phosphosites. Furthermore, it reveals another layer of regulatory diversification since each of the three kinases exhibited selective phosphorylation of the different substrates despite sharing a common core motif.

## Discussion

In the present study we have used quantitative phosphoproteomics to obtain a snapshot of changes that occur in protein phosphorylation during $Ca^{2+}$-dependent exocytosis of synaptic vesicles. A surprisingly complex pattern of both up- and downregulated sites was observed, frequently within the same protein, with the proteins of the active zone being most conspicuous.

Synaptosomes are uniquely suitable for the study of phosphorylation-dephosphorylation events that are associated with synaptic vesicle cycling. Although even the most highly enriched fraction of nerve terminals still contains significant contamination from many other parts of the starting tissue (as confirmed by the composition of the proteome of this fraction), synaptosomes represent the only compartment in the extract that forms membrane enclosed structures retaining basic cellular properties. These include the ability of generating and maintaining a membrane potential and of synthesizing ATP from externally added glucose or ketone bodies (*Nicholls, 2003*; *Choi et al., 2011*). Thus, physiological ATP-levels can only be maintained inside nerve terminals whereas residual external ATP is likely to be rapidly degraded by ATPases, in particular by uncoupled ion pumps. Consequently, protein kinases can only operate inside the nerve terminal during the experiment. Indeed, our comparison of the phosphoproteome versus the total proteome of the synaptosome fraction revealed a significant enrichment of proteins involved in presynaptic function while other proteins (including those of the postsynapse) were reduced or conspicuously absent (*Figure 2—figure supplement 2*).

Although our analysis of phosphosites is certainly not yet comprehensive, several important conclusions can be drawn from our data. First, only a limited number of phosphosites is regulated by stimulation whereas the majority of the quantified sites did not change. Second, almost an identical pattern of regulated sites was observed regardless of whether the $Ca^{2+}$-free control was depolarized or not. This agrees well with the observation that depolarization in the absence of $Ca^{2+}$ resulted only in very limited changes of the phosphosite pattern. This is remarkable when considering that clamp-depolarization by potassium is nonphysiological and may have deleterious side effects (e.g. drop in ATP levels). Consequently, our data show that the phosphorylation changes observed after stimulation are dominated by $Ca^{2+}$-dependent kinases (such as CaMKII, PKC) and phosphatases (such as calcineurin), all of which are implicated in regulating presynaptic function (*Scholz and Palfrey, 1998*; *Brager et al., 2003*; *Leenders and Sheng, 2005*).

In addition, our data hint at the activation of a network of kinases and phosphatases that operates downstream of the primary $Ca^{2+}$-dependent enzymes. For instance, calcineurin upregulates adenylate cyclase in nerve terminals (*Ferguson and Storm, 2004*; *Chan et al., 2005*), increasing cAMP levels and activating protein kinase A. Also, we observed increased phosphorylation of Ser567 in the regulatory subunit of protein phosphatase 2A (PP2A), which is known to enhance its activity (*Ahn et al., 2007*). Moreover, our motif enrichment analysis suggests that CdK5, ERK1/2 and GSK-3 are downregulated upon stimulation. Another example is contributed by the increased phosphorylation of protein phosphatase 1 regulatory subunit Neurabin-2 at S100 that decreases its activity (*Colbran, 2004*). We conclude that the activity of kinases and phosphatases is precisely coordinated: upon $Ca^{2+}$ entry, CaMKII, PKA, PKC as well as PP2A and PP2B are activated whereas proline directed kinases (e.g. CdK5 and GSK-3) as well as PP1 appear to be downregulated.

The panoply of phosphorylation sites and phosphorylation changes detected in many proteins of the presynaptic active zone was surprising. Active zone proteins are thought to structure the site for synaptic vesicle docking and fusion by forming dynamic scaffolds, and several of these proteins such as RIM and Munc13 have been shown to play a crucial role in vesicle docking and priming (*Südhof, 2012*). Only a few of these phosphorylation sites have been previously studied, such as phosphorylation dependent binding of 14-3-3 adaptor protein to bassoon at Ser2844 or the cAMP-regulated site in RIM1 (Ser413) that plays a role in long-term potentiation (*Lonart et al., 2003*;

*Kaeser et al., 2008*). Similar to the well-characterized and multiply phosphorylated synapsin1, active zone proteins contain sites that exhibit opposing changes in response to $Ca^{2+}$.

The functional significance of such multiple phosphorylation remains to be explored. There are many examples showing that within such proteins individual phosphosites can have specific functions, e.g. in regulating the activity of enzymes such as CaMKII (*Cohen, 2000*; *Coultrap and Bayer, 2012*) or channels such as Kv2.1 (*Park et al., 2006*). At present, we do not yet know whether there are individual sites on a given active zone protein that are solely responsible for a functional change (e.g. in regulating binding to a specific protein) or whether multiple phosphosites cooperate in changing the overall properties of a protein. Multi-site phosphorylation may modulate the binding affinity of a protein to other proteins or block binding by increasing its overall negative charge leading to repulsion from other proteins and charged membrane lipids (*Powell et al., 2000*; *Arbuzova et al., 2002*; *Ferreon et al., 2009*). Also, our data only represent a snapshot of phosphorylation changes 2 min after depolarization, i.e. when exo- and endocytosis are both at elevated steady-state levels. Thus it is conceivable that transient changes occurring in the first seconds after stimulations were missed.

Intriguingly, in our data no significant changes were observed in the phosphorylation status of SNARE and other proteins that are mediating exocytosis of synaptic vesicles (e.g syntaxin1A Thr10, Ser14 (*Rickman and Duncan, 2010*), Ser59, Ser64; VAMP-2 Ser75, Ser80; SNAP-25 Ser25; Munc18 Ser506, Ser507, Ser509, Ser593 and NSF Ser739) (Supplementary file 2, *Kohansal-Nodehi et al., 2016*). Thus it appears that the rapid feedback loops governed by kinases and phosphatases, which are activated during stimulation, do not target the exocytotic machinery itself but rather upstream processes such as docking, priming, and endocytosis. Further detailed studies will be required to unravel the functional consequences of these changes at the molecular level.

## Materials and methods

### Chemicals, reagents and buffers

Sucrose, glucose monohydrate, sodium chloride (NaCl), potassium chloride (KCl), magnesium chloride hexahydrate ($MgCl_2x6H_2O$), sodium hydrogen carbonate ($NaHCO_3$), sodium hydrogen phosphate dihydrate ($Na_2HPO_4x2H_2O$), sodium hydroxide (NaOH) were purchased from Merck KGaA (Darmstadt, Germany). Calcium chloride ($CaCl_2$), nicotinamide adenine dinucleotide phosphate hydrate ($NADPx1H_2O$), ammonium hydrogen carbonate ($NH_4HCO_3$), 2-chloroacetamide (CAA), acetone (Chromasolv grade), glutamate dehydrogenase (GluDH), formic acid (FA), dithiothreitol (DTT), pepstatin-A, trifluoroacetic acid (TFA), Nonidet P40, tetraethylammonium bromide (TEAB), 2,5-dihydroxybenzoic acid (DHB), ammonium formate ($NH_4HCO_2$), Ficoll PM 400, sodium cyanoborohydride ($NaBD_3CN$), formaldehyde ($CH_2O$) were purchased from Sigma-Aldrich (Steinheim, Germany). Pierce bicinchoninic acid protein concentration assay (BCA), albumin standard, Pierce protease and phosphatase inhibitor, acetonitrile (ACN, LC-MS grade) were purchased from Thermo Fisher Scientific (Rockford, IL). 2-[4-(2-hydroxyethyl)piperazin-1-yl]ethanesulfonic acid (Hepes), ethylene glycol tetraacetic acid (EGTA) were purchased from Gerbu Biotechnik (Heidelberg, Germany). The remaining chemicals and reagents were purchased from individual supplier: sequencing grade modified trypsin (Promega, Madison, WI), formaldehyde (D2, 98%, $CD_2O$, Cambridge Isotope Laboratories, Andover, MA), ammonium hydroxide ($NH_4OH$, BAKER ANALYZED, Deventer, the Netherlands), PMSF (AppliChem, Darmstadt, Germany), 2-Amino-2-hydroxymethyl-propane-1,3-diol (Tris, VWR International, Leuven, Belgium), Rapigest (Waters, Milford, MA), titanium dioxide beads ($TiO_2$, GL Sciences Inc., Tokyo, Japan).

Sucrose buffer: 320 mM sucrose, 5 mM Hepes, pH=7.4; sodium buffer: 10 mM glucose, 5 mM KCl, 140 mM NaCl, 5 mM $NaHCO_3$, 1 mM $MgCl_2$, 1.2 mM $Na_2HPO_4$, 20 mM Hepes, pH=7.4; lysis buffer: 50 mM Tris, 150 mM NaCl, 1% Nonidet P40, pH=7.4 containing Pierce protease and phosphatase inhibitors; 6%, 9% and 13% Ficoll solutions were prepared in sucrose buffer (pH=7.4). The buffers were prepared using nanopure water obtained with an electric resistance of greater that 18 MΩ from a Mili-Q purification system (MerckMillipore, Darmstadt, Germany). Solutions for mass spectrometry analysis were prepared using LiChrosolv water (Merck KGaA, Darmstadt, Germany).

## Isolation of synaptosomes and glutamate release assay

Wistar rats originated from the local animal facility were kept until use at a 12:12 hr light/dark cycle with food and water ad libitum. Rats of age between 5 to 6 weeks were sacrificed by cervical dislocation followed by decapitation. The brains were removed from the scull and placed into an ice-cold sucrose buffer. Cerebral cortises and cerebellums were dissected, pooled together and subjected to the synaptosomes enrichment by the non-continuous Ficoll gradient as described previously (*Fischer von Mollard et al., 1991*). Glutamate release was monitored by enzymatic conversion of glutamate as previously reported (*Nicholls and Sihra, 1986*) with minor modifications. The synaptosomal pellet was resuspended in ice cold sucrose buffer with the addition of protease inhibitors (PMSF, 200 mM and Pepstatin-A, 1 µg/ml). For glutamate release assay 2.5 mg of synaptosomes were resuspended in 2.5 ml of sodium buffer and incubated with stirring for 5 min at 37°C. Depending on the stimulation, either $CaCl_2$(1.3 mM) or EGTA (0.5 mM) and NADP (1 mM) was added and incubation was continued for 3 min, then GluDH (200 U) was added and after 3 min KCl (50 mM) was spiked in. Generation of NADPH was monitored by fluorescence for 2 min. After this time synaptosomes were collected for mass spectrometry and western blot analysis.

## Mass spectrometry sample collection

Synaptosomes were spun down at 6200 × g for 2 min in 4°C, resuspended in 100 µl lysis buffer. Then 900 µl of 100 mM TEAB (pH=8) buffer was added. The samples were homogenized (RW20-DZM, IKA, Staufen, Germany) at maximum speed (2000 rpm) by 3 strokes. Global protein concentration was determined by BCA protein concentration assay, using albumin as a standard. 1 mg of proteins per condition was precipitated by acetone (100%) over night at -20°C. Proteins were spun down at 16,000 × g for 30 min at 4°C. The pellets were washed once with 200 µl of 80% acetone and centrifuged again at 16,000 × g for 30 min at 4°C. The protein pellets were dried briefly in the air at room temperature and later stored at -20°C for further analysis.

## Protein digestion

Dried protein pellets were resuspended in 80 µl of 1% Rapigest (in 100 mM TEAB, pH=8) by vortexing and incubated in thermoshaker at 60°C for 15 min at 1050 rpm. Twenty µl of 50 mM DTT (in 100 mM TEAB, pH=8) were added; samples were incubated at 60°C for 45 min 1050 rpm. Then 20 µl of 100 mM CAA (in 100 mM TEAB, pH=8) were added; samples were incubated in thermoshaker at 37°C for 30 min at 750 rpm. Samples were diluted by 100 mM TEAB (pH=8) to decrease final concentration of Rapigest to 0.1% and then 1:20 trypsin to protein ratio was added. Samples were digested overnight at 37°C, at 750 rpm. The digestion was stopped by addition of 4 µl of 100% FA and incubation for 2 hr at 37°C at 750 rpm. Samples were spun down at 13,000 rpm for 30 min at 4°C; supernatant was transferred to the new vials, dried in vacuum concentrator and kept in -20°C for further analysis.

For determination of the proteome 50 µg of synaptosomes were separated by SDS-PAGE using a 4–12% gradient gel (NuPAGE, Life Technologies, Carlsbad, CA) according to Shevchenko et al (*Shevchenko et al., 1996*). For each sample, three biological replicates were carried out. Each lane was cut into 23 pieces. Proteins in each piece were in-gel digested overnight by trypsin and later digested peptides were extracted by 20 µl water followed by addition of 80 µl 100% acetonitrile. Extracted peptides were acidified by formic acid to final concentration of 0.05% vol/vol, dried in the vacuum concentrator and analyzed by liquid chromatography (LC) coupled to mass spectrometry (MS/MS).

## Stable isotope dimethyl labeling

Stable isotope dimethyl labeling was performed according to Boersema *et al.* (*Boersema et al., 2009*). Briefly, digested sample were labeled separately to yield light and heavy isotope tags by adding 4 µl of 4% (vol/vol) $CH_2O$ followed by 4 µl of 4% (vol/vol) $CH_2O$ and $CD_2O$, to light and heavy samples respectively. Later 4 µl of 0.6 M $NaBD_3CN$ and 16 µl of 1% (vol/vol) ammonia and 8 µl of 5% FA (vol/vol) were added to the labeling reaction of both heavy and light samples. Light and heavy samples were mixed at a volume ratio of 1:1 according to experimental design (*Figure 2—figure supplement 1*) and were kept on ice for further analysis.

## Strong cationic exchange chromatography

Mixed light and heavy labeled peptides were diluted to a final volume of 1 ml with solvent A (10 mM $NH_4HCO_2$, 30% ACN (v/v), pH=2.7) and loaded onto a Mono S column PC 1.6/5 (Pharmacia Biotech, Uppsala, Sweden) at a flow rate of 100 µl/min. Elution was performed with a gradient of 0–90% solvent B (500 mM $NH_4HCO_2$, 30% ACN ((v/v), pH=2.7) over 40 min. The first twelve fractions (0.2 ml) including the flow-through were collected and subjected separately to phosphopeptide enrichment.

## Phosphopeptide enrichment

Phosphopeptides were enriched using $TiO_2$ chromatography as previously described by Larsen et al (*Larsen et al., 2005*). Digested peptides were dissolved in 60 µl of 200 mg/ml DHB in 80% ACN, 5% TFA and loaded onto $TiO_2$ columns. The columns were washed three times with 60 µl of 200 mg/ml DHB in 80% ACN, 5% TFA and five times with 60 µl of 80% ACN, 5% TFA. Then, phosphopeptides were eluted by three consecutive additions of 40 µl of 0.3 N $NH_4OH$, pH $\geq$ 10.5. Eluted phospho-peptides were dried in the vacuum concentrator for further MS analysis.

## Mass spectrometry analysis

Enriched phosphopeptides were analyzed on a Q-Exactive hybrid Quadrupole-Orbitrap mass spectrometer (Thermo Fisher Scientific, Dreieich, Germany) coupled to a NanoLC pump (EASY-nLC, Thermo Fisher Scientific, Dreieich). The peptides were pre-concentrated on a Reversed Phase-C18 precolumn (0.15 mm ID × 20 mm self-packed with Reprosil-Pur 120 C18-AQ 5 µm, Dr. Maisch GmbH, Ammerbuch-Entringen, Germany) and then separated by reversed phase-C18 nanoflow chromatography (0.075 mm ID × 200 mm self-packed with Reprosil-Pur 120 C18-AQ, 3 µm, Dr. Maisch GmbH). Peptides were injected with solvent A (0.1% FA, 5% acetonitrile in water) at a flow rate 320 nL/min and eluted by 0–37% solvent B (80% acetonitrile, 0.1% FA in water) with an overall run-time of 45 min. Separated peptides were ionized by electrospray ionization (ESI) source in a positive ion mode. Full scan MS spectra were acquired in the range of 350–1600 m/z at a resolution of 70,000. The top 12 most intense peaks from the survey scan were selected for fragmentation with Higher-energy Collisional Dissociation (HCD, 28 of normalized collision energy).

For proteomics analysis, LTQ Orbitrap XL (Thermo Fisher Scientific) coupled to Agilent 1100-LC system (Agilent Technologies, Waldbronn, Germany) was used. Peptides were loaded onto a trap column packed in-house (0.15 mm ID × 20 mm self-packed with Reprosil-Pur 120 C18-AQ 5 µm, Dr. Maisch GmbH) and separated at a flow rate of 15 µl/min on an analytical column (0.075 mm ID × 120 mm self-packed with Reprosil-Pur 120 C18-AQ, 3 µm, Dr. Maisch GmbH). Peptide were eluted from the column with 3–35% solvent B with an overall run-time of 45 min. Separated peptides were ionized by electrospray ionization (ESI) source in a positive ion mode. Full scan MS spectra were acquired in the range of 350–1600 m/z at a resolution of 35,000. The top 10 peaks from survey scan were selected for fragmentation with Collision-Induced Dissociation (CID) with 37.5 normalized collision energy.

## Data processing and bioinformatics analysis

Acquired MS spectra were processed using the MaxQuant software package version 1.5.0.25 (*Cox and Mann, 2008*). Spectra were searched using the Andromeda search engine (*Cox et al., 2011*) against the proteome database of Rattus Norvegicus (Uniprot complete proteome updated at 2015-01-07, with 29,378 entries). MaxQuant search was configured as follows; the mass tolerance was set to 20 and 4.5 ppm for first and main peptide search, respectively. The multiplicity was set to two (DimethLys0 and DimethNter0 was for 'light' and DimethLys4 and DimethNter4 for 'heavy' samples). Trypsin/P was fixed as protease and maximum of 2 missed cleavages were allowed. Carbami-domethylation of cysteine was set as fixed modification and methionine oxidation, acetylation (on N-terminal), phosphorylation on serine/threonine/tyrosine were specified as variable modifications. False discovery rate of 1% was applied and re-quantification was enabled. For proteomics analysis MaxQuant search was configured like phosphoproteomics search except that multiplicity was set to one with methionine oxidation and acetylation (on N-terminal) as variable modifications. The phos-pho (STY) output file from the MaxQuant was processed by 'Perseus version 1.5.0.9' (*Cox and Mann, 2008*) for downstream data analysis. In each comparison phosphosites quantified in at least

two out of three biological replicates were selected for further analysis. Perseus data processing was configured as follows; reverse hits were removed and minimum localization probability of 0.75 was set as criterion to accept phosphosites for further analysis. The normalized ratios and the intensities of phosphosites reported by MaxQuant were $\log_2$ and $\log_{10}$ transformed respectively, and were used to plot the distribution of quantified phosphosites (*Figure 2A,B*). The significantly regulated phosphosites (significant outliers) were determined by two-tailed 'Significant A' test ($p$-value $\leq 0.05$) using the Perseus software (*Cox and Mann, 2008*; *Cox et al., 2009*).

The Motif-X tool (*Schwartz and Gygi, 2005*) was used remotely from motif-x.med.harvard.edu/motif-x.html for sequence motif analysis. Sequences of upregulated or downregulated phosphorylation sites in the comparisons were independently searched against the IPI Rat Proteome as background. The default suggested setting (occurrences = 20 and significance = 0.000001) were used for the analysis. Central character was set as 'S' and sequence length was specified as 31.

The Ingenuity Pathway Analyses software (IPA, build version 21901358, Qiagen, Redwood City, CA) was used to identify significantly enriched biological functions in the synaptosomal proteome and phosphoproteome data sets. The comparison analysis function of IPA was then used to further determine which functional clusters differ significantly between the two datasets. Only clusters having a p-value < 0.01 were considered for the analysis. Right-tailed Fisher's exact test was used to calculate a p-value determining the probability that each biological function assigned to that data set is due to chance alone. Functionally redundant clusters were manually removed. The heat map was generated using Microsoft Excel.

## Kinase assay

*In vitro* kinase assays were carried out using the ADP-Glo kinase assay kit (Promega, Meinheim, Germany) in three replicates. In all assays 0.2 µg/µl of each peptide were used. First, standard curves were obtained by making a series of ADP-ATP (provided together with kinase assay kit) dilutions based on the kit protocol, and the luminescence of each dilution was measured. Then, we optimized the concentration of each kinase and of the ATP concentration to achieve 100% conversion of ATP to ADP based on the standard curves, defined as 100% activity. Then the phosphorylation of each investigated phosphosite by the respective kinase was calculated and reported as percent of activity of the enzyme. For PKA (Promega, Meinheim, Germany) assay, Kemptide (provided with the enzyme) was used as positive control peptide, and 5 µM ATP and 0.7 U/µl enzyme were used. For CaMKII (New England Biolabs, Frankfurt, Germany) we first activated the enzyme by addition of 5 µM ATP, 1.2 µM calmodulin (provided with the enzyme) and 2 mM $CaCl_2$ and 30 min incubation at 37°C. Then, for the kinase reactions we used 5 µM ATP and 12 U/µl of kinase. Autocamide (GenScript, Piscataway, NJ) was used as positive control peptide. For PKC (Promega, Meinheim, Germany) assays, Neurogranin (Provided with enzyme) was used as positive control peptide, and the incubation was carried out with 25 µM ATP and 5 ng/µl enzyme. Peptides were purchased from commercial supplier (Thermo Fisher Scientific, Ulm, Germany). The sequences of the peptides are listed in *Figure 6—source data 1*.

## Immunoblotting

10 µg of synaptosomal extract was used for immunoblotting according to standard procedures (*Liu et al., 2014*) after separation of samples by 4–12% gradient SDS-PAGE gels. The following primary antibodies were used: synapsin1, dynamin1-3 and α-tubulin (Synaptic Systems, Göttingen, Germany), phospho S774-dynamin1, phospho S603-synapsin1 (Rockland, Pottstow, PA). All primary antibodies were used at a dilution of 1:1000. HRP-conjugated mouse, rat or sheep secondary antibodies were used for detection of the respective primary antibodies. Immunoreactive proteins were developed with enhanced chemiluminescence reagent (SuperSignal West Pico kit, Thermo Fisher Scientific, Rockford, IL), and signals were detected with an LAS-1000 imaging system (Fujifilm, Tokyo, Japan). Quantification of immunoblots was performed using ImageJ, and Prism (version 5.04, GraphPad software) was used for the statistical analysis.

## Acknowledgements

The authors thank Momchil Ninov and Ángel Pérez-Lara for help with the preparation of synaptosomes and with the strong cation exchange chromatography, respectively, and Uwe Pleßmann,

Monika Raabe, and Annika Kühn for technical support in mass spectrometry. Also we thank Jasmin Corso for helpful discussions and Kuan-Ting Pan for critically reviewing the manuscript. This work was supported by grants from the EU Commission (SynSys, Eurospin) to RJ and the stipend from the International Max Planck Research School (IMPRS) for Molecular Biology of Göttingen to MK.

## Additional information

### Competing interests
RJ: Reviewing editor, *eLife*. The other authors declare that no competing interests exist.

### Funding

| Funder | Grant reference number | Author |
|---|---|---|
| European Commission | grant agreement number: 242167 | Reinhard Jahn |
| Max Planck Research School (IMPRS) for Molecular Biology | | Mahdokht Kohansal-Nodehi |

The funders had no role in study design, data collection and interpretation, or the decision to submit the work for publication.

### Author contributions
MK-N, DC, Conception and design, Acquisition of data, Analysis and interpretation of data, Drafting or revising the article; JJEC, Analysis and interpretation of IPA data, Revising the article; HU, Interpretation of data, Revising the article, Conception and design; RJ, Interpretation of data, Conception and design, Drafting or revising the article

### Author ORCIDs
Mahdokht Kohansal-Nodehi, http://orcid.org/0000-0002-3898-5197

John JE Chua, http://orcid.org/0000-0002-5615-1014

Reinhard Jahn, http://orcid.org/0000-0003-1542-3498

Dominika Czernik, http://orcid.org/0000-0001-7911-0867

### Ethics
Animal experimentation: All animal procedures used here fully comply with the guidelines as stipulated in the section 4 of the Animal Welfare Law of the Federal Republic of Germany (section 4 of TierSchG, Tierschutzgesetz der Bundesrepublik Deutschland). All procedures were performed in the animal facility at the Max-Planck-Institute for Biophysical Chemistry, Göttingen, Germany registered accordingly to the section 11 Abs. 1 TierSchG as documented by 39 20 00_2a Si/rö, dated 11th Dec 2013 ("Erlaubnis, Wirbeltiere zur Versuchszwecken zu züchten und zu halten"), by the Veterinär- und Verbraucherschutzamt für den Landkreis und die Stadt Göttingen and examined regularly by the supervisory veterinary authority of the Landkreis Göttingen. All procedures were supervised by the animal welfare officer and the animal welfare committee of the Max-Planck-Institute for Biophysical Chemistry, Göttingen, Germany established accordingly to the TierSchG and the regulation about animal used in experiments, dated on 1st Aug 2013 (TierSchVersV, Tierschutz-Versuchstier-Verordnung).

## Additional files

### Major datasets
The following dataset was generated:

| Author(s) | Year | Dataset title | Dataset URL | Database, license, and accessibility information |
|---|---|---|---|---|
| Kohansal-Nodehi M, Chua J, Urlaub H, Jahn R, Czernik D | 2016 | Data from: Analysis of protein phosphorylation in nerve terminal reveals extensive changes in active zone proteins upon exocytosis | http://dx.doi.org/10.5061/dryad.dh371 | Publicly available at Dryad Digital Repository under a CC0 Public Domain Dedication |

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
