## [Decision Letter]

Thank you for submitting your article "Analysis of protein phosphorylation in nerve terminal reveals extensive changes in active zone proteins upon exocytosis" for consideration by eLife. Your article has been favorably evaluated by a Senior editor and three reviewers, one of whom, Mary B. Kennedy, is a member of our Board of Reviewing Editors.

The reviewers have discussed the reviews with one another and the Reviewing Editor has drafted this decision to help you prepare a revised submission.

Summary:

The study by Kohansal-Nodehi et al. couples classical methods of studying the biochemistry of nerve terminals with "modern," mass-spec driven, proteomic studies. The presentation makes clear how proteomic studies can be analyzed to clarify and extend functional mechanistic studies carried out in more directed experiments. The study reveals two new interesting regulatory phenomena. The active zone proteins piccolo and bassoon are more extensively regulated by depolarization of synaptosomes in the presence of Ca^2+^ than predicted from earlier studies. The study also reveals that phosphosites that are up-regulated by depolarization plus Ca^2+^ are dominated by sequence motifs that are recognized by CaMKII, PKA, and PKC, as might be predicted; however, phosphosites that are down-regulated are dominated by the SP motif recognized by ERK1/2, CdK5, or CdK5, suggesting that phosphatases activated by depolarization and Ca^2+^ act predominantly on sites regulated by the latter kinases, at least in the first few minutes after stimulation.

The authors analyzed the phosphoproteome of rat synaptosomes, comparing preparations that were depolarized with KCL (clamped depolarization) in the presence of calcium ("stimulated" condition) to preparations that were either not depolarized, or depolarized in the absence of calcium. The authors identify up- or down-regulation of the phosphorylation status of a total of 252 sites in (mostly) pre-synaptic proteins. They categorize the regulated phosphosites based on protein function and provide an overview of the changes in the phosphorylation status of a selection of active zone proteins, synaptic vesicle proteins, plasma membrane proteins, and proteins involved in Clathrin mediated exocytosis. The Authors used standard omics procedures to label proteins in the two conditions with H/D using dimethylation, they enriched phosphopeptides using a TiO_2_ trap.

Overall, their analysis of presynaptic, stimulation-dependent phosphorylation changes provides a valuable overview that will be of interest to many researchers in the presynaptic field. The datasets should be extremely useful for more detailed analyses of the functional consequences and proximate causes of individual, activity-dependent phosphorylation or de-phosphorylation events. The study provides a more global context in which studies of the functions and regulation of individual proteins can be designed and interpreted. It paves the way for a new paradigm for studies of the dynamics of protein regulation that underlie control of transmitter release, and recovery of the terminal from a round of stimulated release.

Essential revisions:

1) One reviewer felt that the stimulated condition that the Authors have chosen is somewhat non-physiological, i.e. a clamped depolarization with the membrane potential close to zero. It is a situation in which the status of voltage-dependent ion channels and ion pumps is far from the normal electrical activity. A close-threshold depolarization with 4-aminopyridine would have been a more physiological condition to perform the study. This reviewer felt that the terminals would be subjected to a calcium overload. In contrast, another reviewer felt that this is not a problem because, with the K^+^ stimulus, inactivation of voltage-dependent calcium channels would limit large calcium influx to the first few seconds, and that ATP driven calcium pumps and the sodium/calcium exchanger would not be strongly affected by the clamped depolarization as they are not largely voltage driven. It would be useful for the authors to comment on these points in the manuscript.

2) In the Discussion, the authors state that "it appears that the rapid feedback loops governed by kinases and phosphatases, which are activated during stimulation, do not target the exocytotic machinery itself but rather upstream processes such as docking, priming and endocytosis." However, the authors have captured a snapshot at 2 min after stimulus. Since the turnover number of most protein kinases is around 1/sec, and the turnover numbers of protein phosphatases can be much faster, the authors could have missed even more rapid cycles that occur in the first 10, 30, or 60 s. It would be better to add the statement that regulatory cycles of the exocytotic machinery may be complete by 2 min after the stimulus, and thus more rapid quenching of the reactions before analysis might be required to capture them.

3). The authors should explain in more detail why the correlation of the ratios of the 3 biological replicates (Figure 2—figure supplement 4) is low for their two "control" conditions (K^+^, EGTA vs. EGTA), and state whether or not this affects the pairwise comparison to the stimulated condition (K^+^, Ca^2+^).

4) Because *eLife* is a journal with a broad readership, it would be appropriate to include an example of the raw data from which the extent of phosphorylation of a site in each sample is determined. Perhaps the authors could include an additional Figure 2 supplement that shows the light and heavy peaks for S603-synapsin1 and S744-dynamin1 from each sample, with an indication of which fraction of the SCX fractionation these peaks were found in.

5) The concepts behind the "Data processing and bioinformatics analysis" section, i.e. how the raw data were converted into the figure panels, should be explained in more detail – especially for Figure 2 and its supplements.

6) The authors suggest that the kinases targeting the SP motif (CdK5, ERK1/2, GSK-3) are down-regulated in response to stimulation and/or that the turnover at these sites may be faster. This would indeed be very interesting, have the authors attempted to verify either of these possibilities? It seems more likely that calcium-stimulated phosphatases are stimulated to dephosphorylate these sites at a faster rate; whereas, the kinases are not stimulated by calcium. Therefore, dephosphorylation dominates after the calcium stimulus. Please comment.

---

## [Author Response]

Essential revisions:

1) One reviewer felt that the stimulated condition that the Authors have chosen is somewhat non-physiological, i.e. a clamped depolarization with the membrane potential close to zero. It is a situation in which the status of voltage-dependent ion channels and ion pumps is far from the normal electrical activity. A close-threshold depolarization with 4-aminopyridine would have been a more physiological condition to perform the study. This reviewer felt that the terminals would be subjected to a calcium overload. In contrast, another reviewer felt that this is not a problem because, with the K^+^ stimulus, inactivation of voltage-dependent calcium channels would limit large calcium influx to the first few seconds, and that ATP driven calcium pumps and the sodium/calcium exchanger would not be strongly affected by the clamped depolarization as they are not largely voltage driven. It would be useful for the authors to comment on these points in the manuscript.

We agree with the reviewer that KCl stimulation leads to a clamped depolarization and thus is far from a physiological stimulus. This, however, is inherent to the entire approach: synaptosomes (pinched-off nerve terminals) are not suitable for fine-tuning of synaptic excitation. On the other hand, synaptosomes have served since decades as a convenient and biochemically accessible model system for studying basic features of the exo- and endocytotic machinery. In fact, a lot of basic knowledge about the functioning of vesicle release was first obtained from studies on synaptosomes that were stimulated by KCl-depolarization (including phosphorylation and dephosphorylation of key proteins such as dynamin and synapsin). Many previous papers have shown that exocytotic release persists for several minutes after KCl depolarization (see e.g. [Nicholls and Sihra, 1986]), and that calcium influx ceases after an initial burst (McMahon and Nicholls, 1991), see also other papers by D.W. Nicholls and coworkers in the eighties and nineties of last century where this preparation was extensively characterized.

We acknowledge that potassium channel blockers were used previously to drive calcium-dependent exocytosis, triggering depolarization via firing of voltage-gated sodium channels. However, this type of activation is less reproducible and less well controlled experimentally. We have now acknowledged these inherent limitations more explicitly (see section “Phosphoproteomics of isolated nerve terminals under different stimulation conditions”, first, second and third paragraphs) but for the reasons summarized above we consider our experimental approach as conclusive despite its limitations.

*2) In the Discussion, the authors state that "it appears that the rapid feedback loops governed by kinases and phosphatases, which are activated during stimulation, do not target the exocytotic machinery itself but rather upstream processes such as docking, priming and endocytosis." However, the authors have captured a snapshot at 2 min after stimulus. Since the turnover number of most protein kinases is around 1/sec, and the turnover numbers of protein phosphatases can be much faster, the authors could have missed even more rapid cycles that occur in the first 10, 30, or 60 s. It would be better to add the statement that regulatory cycles of the exocytotic machinery may be complete by 2 min after the stimulus, and thus more rapid quenching of the reactions before analysis might be required to capture them.*

We agree with the reviewers that we are certainly missing fast and reversible changes, and shorter time intervals after stimulation will need to be analyzed in the future. On the other hand, exocytotic release of glutamate continues for minutes (see above), and thus we are probably capturing an activated steady-state status of the exo- and endocytotic machinery. We have added a paragraph in the Discussion to clarify this point in more detail (sixth paragraph).

3) The authors should explain in more detail why the correlation of the ratios of the 3 biological replicates (Figure 2—figure supplement 4) is low for their two "control" conditions (K^+^, EGTA vs. EGTA), and state whether or not this affects the pairwise comparison to the stimulated condition (K^+^, Ca^2+^).

Low correlation is observable only when correlating the ratios between the stimulated in the presence of EGTA (K^+^, EGTA) and non-stimulated phosphosites (EGTA). In contrast, an excellent correlation is observed when intensities (sum of intensities in both conditions, labeled as heavy and light) are correlated (Figure 2—figure supplement 4). As further evidence for the high reproducibility between the replicates in these two conditions, we plotted the intensity values separately in K^+^, EGTA (heavy) and EGTA (light) conditions and calculated the Pearson correlation coefficient (Figure 7).

Author response image 1.Correlation of phosphosites between biological replicates.Correlation plots depicting correlation of phosphosites between three biological replicates at Log_10_ intensity level in depolarization without Ca^2+^ (K^+^, EGTA, heavy) condition (**A**), no depolarization (EGTA, light) condition (**B**).**DOI:**
http://dx.doi.org/10.7554/eLife.14530.016

The low correlation in the ratio level is simply a reflection of the fact that there is no major change of the phosphorylation status if the synaptosomes are depolarized in the absence of extracellular calcium. Consequently, most ratios cluster close to 1 (log2 scale 0, Figure 2—figure supplement 4). In contrast, upon depolarization in the presence of calcium (K^+^, Ca^2+^) many phosphosites are either up- or downregulated, which results in a high correlation not only of the intensity levels but also in the ratios (Figure 2—figure supplement 3).

4) Because eLife is a journal with a broad readership, it would be appropriate to include an example of the raw data from which the extent of phosphorylation of a site in each sample is determined. Perhaps the authors could include an additional Figure 2 supplement that shows the light and heavy peaks for S603-synapsin1 and S744-dynamin1 from each sample, with an indication of which fraction of the SCX fractionation these peaks were found in.

As suggested we have now included a new supplemental figure showing MS1 spectra of the synapsin Ser603 and dynamin Ser774 in light (K^+^, Ca^2+^) and heavy (K^+^, EGTA) isotopic patterns (Figure 2—figure supplement 5) together with annotated MS/MS spectra for synapsin Ser603 and dynamin Ser774 phosphosites (Figure 2—figure supplement 5).

5) The concepts behind the "Data processing and bioinformatics analysis" section, i.e. how the raw data were converted into the figure panels, should be explained in more detail – especially for Figure 2 and its supplements.

As suggested we have added more details to the section "Data processing and bioinformatics analysis" to make sure that the procedures chosen for the analysis of the data is clear for readers (first paragraph).

6) The authors suggest that the kinases targeting the SP motif (CdK5, ERK1/2, GSK-3) are down-regulated in response to stimulation and/or that the turnover at these sites may be faster. This would indeed be very interesting, have the authors attempted to verify either of these possibilities? It seems more likely that calcium-stimulated phosphatases are stimulated to dephosphorylate these sites at a faster rate; whereas, the kinases are not stimulated by calcium. Therefore, dephosphorylation dominates after the calcium stimulus. Please comment.

Our data indicate that both of the suggested mechanisms play a role: For ERK1/2 we consider downregulation as unlikely since two phosphosites in the active loop of ERK1/2 (T183, Y185, Supplementary file 2, K^+^, EGTA vs K^+^, Ca^2+^) were found to be upregulated. It is known that upregulation of these sites increases the activity of ERK1/2 kinases (this point is mentioned now in the second paragraph of the section “Calcium-dependent kinases and phosphatases are mainly responsible for phosphorylation changes after stimulation”). In the case of GSK3, the enzyme is probably inactivated during stimulation, for the following reasons: First, two phosphorylation sites (S278, Y279) known to result in activation of the kinase (Liang and Chuang, 2007) were downregulated upon stimulation. Second, it was shown previously that upon stimulation, an inhibitory site of the GSK-3 (S9) is phosphorylated by CaMKII and inactivates the kinase (Song et al., 2010). With respect to CdK5, our data show that phosphosites known to be phosphorylated by CdK5 are downregulated upon stimulation (e.g. dynamin1 S774, septin5 S17). On the other hand, PP2A is probably activated upon stimulation (phosphorylation of Ser567 in the regulatory subunit). It is also well stablished that protein phosphatase 2B is activated by high concentration of Ca^2+^.